# Making Open Scholarship More Equitable and Inclusive

**Paul Longley Arthur** [1,*], **Lydia Hearn** [1], **John C. Ryan** [2], **Nirmala Menon** [3] and **Langa Khumalo** [4]

1 School of Arts and Humanities, Edith Cowan University, Mt Lawley, Perth, WA 6050, Australia; l.hearn@ecu.edu.au
2 School of Arts and Social Sciences, Southern Cross University, East Lismore, NSW 2480, Australia; john.c.ryan@scu.edu.au
3 School of Humanities and Social Sciences, Indian Institute of Technology Indore, Indore 453552, India; nmenon@iiti.ac.in
4 South African Centre for Digital Language Resources (SADiLaR), North-West University, Potchefstroom 2520, South Africa; langa.khumalo@nwu.ac.za
* Correspondence: paul.arthur@ecu.edu.au

**Abstract:** Democratizing access to information is an enabler for our digital future. It can transform how knowledge is created, preserved, and shared, and strengthen the connection between academics and the communities they serve. Yet, open scholarship is influenced by history and politics. This article explores the foundations underlying open scholarship as a quest for more just, equitable, and inclusive societies. It analyzes the origins of the open scholarship movement and explores how systemic factors have impacted equality and equity of knowledge access and production according to location, nationality, race, age, gender, and socio-economic circumstances. It highlights how the privileges of the global North permeate academic and technical standards, norms, and infrastructures. It also reviews how the collective design of more open and collaborative networks can engage a richer diversity of communities, enabling greater social inclusion, and presents key examples. By fostering dialogue with multiple stakeholders, more effective avenues for knowledge production and representation can be built based on approaches that are accessible, participatory, interactive, ethical, and transparent, and that reach a far broader public. This expansive vision of open science will lead to a more unified knowledge economy.

**Keywords:** open access; open science; open scholarship; gender; equity; equality; language; justice; environment





## 1. Introduction

A fundamental feature driving open science—or open scholarship—has been the moral call for more just, equitable, and inclusive societies. As enshrined in Article 27 of the United Nations Universal Declaration on Human Rights, "Everyone has the right to freely participate in the cultural life of the community, to enjoy the arts, and to share in scientific advancement and its benefits" [1]. By enhancing visibility and increasing access to scholarly outputs, openness has the potential to foster a wider culture of education and literacy, directly influencing public policy through greater public engagement in new ideas and technology that can enrich the lives and livelihoods of people everywhere [2]. Yet, as declared in the 2021 UNESCO Recommendation on Open Science, "a global understanding of the meaning, opportunities, and challenges of Open Science is still missing" [3]. While the UNESCO statement recognizes the collaborative and inclusive characteristics of open science for knowledge democratization, achieving this requires not only access to open knowledge, open infrastructures, and open communication, but also the open engagement of all societal actors, including those beyond the traditional scientific society, through open dialogue with other knowledge systems to address existing systemic inequalities and to guide scientific work toward solving the most significant human challenges [3].

Despite the opportunities provided by new digital technologies, they are unequally distributed and give rise to a knowledge divide for much of the world. Questions emerge around whether the benefits of open scholarship can be truly democratic, unrestricted, and fully inclusive. How can we unlock access to data, information, and knowledge to yield positive outcomes for all, regardless of socio-economic, gender, geographical, and cultural factors? How is it possible to prevent further inequality between and within communities and the global North–South divide? While open scholarship is an international movement with the potential to provide substantial benefits for universities, businesses, governments, and non-governmental organizations across the world, it also involves a complex array of power relations that may not always be consistent with the goal of inclusive, equitable development [2]. Open scholarship policies, technologies, standards, and models have stemmed primarily from the global North and been applied to the global South, creating new categories of exclusion, with the risk of exacerbating the legacies of colonialist systems of scholarly communication and further disadvantaging the needs or aspirations of diverse, marginalized groups [4]. Initiatives are underway to scale up international collaboration for more transparent, equitable cooperation toward openness to address persistent tensions between those managing scholarly outputs and developing data, tools, software, publications, and workflows, and those promoting an open knowledge environment. Only through greater inclusivity can we reconfigure power relations and involve diverse peoples and populations in knowledge creation, use, and re-use processes [5]. Open scholarship in the humanities can play a central role in this progression.

This article examines the historical and cultural roots of the open scholarship movement to explore the socio-political and technological environment in which it operates and to illustrate how over-representation by the global North has replicated some of the very power imbalances that the movement sought to overcome. Through a narrative review, it analyzes how the foundational values of equality and inclusion have become diluted by an emphasis on the pragmatics of open access to research outputs, data, and educational resources and their transparency, accountability, and excellence rather than on their underlying philosophies and implicit values and ethics. It considers the main barriers and issues restricting equity in open scholarship, focusing locally, nationally, and internationally. It then explores how these problems have been shaped by colonialist and capitalist interests and how, by acknowledging these barriers, we can begin to address the key features of openness: common understanding, common values, common directions, and common goals. Finally, the article delves into possible solutions aimed at encouraging cultural diversity, equity, and inclusion through fostering dialogue with multiple stakeholders to build promising pathways for open scholarship focused on the principles of openness and justice.

Referring to a series of case studies, this article's methodology introduces the reader to and investigates ways in which the humanities have focused on addressing struggles for openness in the global South, including data sovereignty among Indigenous groups. By centering open scholarship socially and culturally, through developing collaborative North–South and South–South networks, and by retaining community ownership of local and Indigenous knowledges, barriers can be removed and learning shared to enrich education more openly and equitably [6]. Examples include (1) the creation of new infrastructure providing language resources for the multiple recognized languages of South Africa, thereby opening access to communities that have been under-resourced and marginalized; (2) the digitization of minority literature to reach the numerous socio-cultural and language communities within India; and (3) aural-digital storytelling—audio-recorded soundwalks, multi-modal sound maps, and crowdsourcing techniques—to engage Indigenous audiences together with academics as well as community groups to grapple with the direct human impacts of climate change, ecosystem degradation, and related urgencies, from the perspective of the humanities.

## 2. The Historical, Political, and Socio-Economic Roots of Open Scholarship

The open scholarship movement is intrinsically linked with the development of the Internet, which brought with it a future of global connectivity and practically unlimited access to and sharing of information and knowledge [2,7]. Open scholarship had its origins prior to the advent of the Internet. Yet, in the 1990s, at the time digital technologies were signaling the prospect that online access to research and publications would be cheaper and faster than ever before, with printed articles becoming increasingly redundant, lucrative publishers were gaining power and control over the purchase costs and dissemination of these outputs [8]. To align with this changing world, a consortia of commercial for-profit publishing companies developed collective "Big Deals", offering university libraries multi-year digital packages including subscriptions to a large market share of their serial journals and instant virtual access to a much greater range of literature, while also making this information more discoverable [9,10]. However, these modifications came at a high cost for academic libraries and research institutions, leading to widespread dissatisfaction with the expensive traditional publishing model [11,12]. To address this, in 1998, the Scholarly Publishing and Academic Resources Coalition (SPARC), an alliance of academic libraries and other organizations, was established to seek alternatives, arguing that if society funds science through paying taxes that support academic salaries and research, then the general public must have access to its findings. This led to a range of key initiatives, including the launch of the open-source Open Journal Systems (OJS) platform, the creation of institutional repositories, and the establishment of the Open Archives Initiative (OAI), all of which were followed from 2002 by a series of major international statements calling for unrestricted online access to research outputs (Budapest Open Access Initiative 2002, Berlin Declaration 2003).

The open scholarship movement has sought to make scholarly communication and research results freely available through the support of a nonprofit ecosystem whose co-operative work promotes a more sustainable, inclusive, and participatory "knowledge commons" [13–17]. Accordingly, since its emergence at the start of the 21st century, many statements have advocated for greater equity in access to publications, open data, copyright revision, open educational resources, open government data, alternative metrics and assessment, and responsive research and development, with some calling for the implementation of Findable, Accessible, Interoperable and Reusable (FAIR) practices to facilitate greater public access, collaboration, and multi-stakeholder and citizen engagement [18–22]. For example, the 2005 Salvador Declaration on Open Access highlighted the importance of open access in promoting equity, envisaging outcomes commensurate with the United Nations' Millennium Development Goals, including large-scale collaborative partnerships for more equitable access to information, especially among developing countries.

Yet, two decades after the foundational Budapest Open Access Initiative (2002) was first drafted and optimism that the Internet would assist in developing and supporting sustainable "knowledge societies," the opportunities are unequally distributed [23]. Despite drives for greater equity of access to and engagement with education through networked platforms and repositories to bridge social divides [24], restricted Internet access, limited digital literacy, the primacy of English language publishing, and crucially, the cost of making research openly available, have become barriers reinforcing the global North–South imbalance and inequities within and across borders [25]. This further highlights the digital divide in ways that impact not just access to technology but also the growth of knowledge and literacy.

Inclusion, diversity, and justice need to be principle guiding factors in the development of the global open scholarship ecosystem. Yet, the links between openness and inclusion are not straightforward. Often, openness and inclusion have been enabled along gender, class, age, religion, ethnicity, education, ability, and geographic lines—and with them a set of material and symbolic barriers that intersect with colonialist and racialist frameworks that still underpin histories and cultural heritage, and limit participation or even representation in the design and implementation of open practices [10,26,27]. Historically marginalized Indigenous and minority populations in both urban areas and regions tend to have lower

levels of digital inclusion. Therefore, openness should not just be about addressing technical problems to make information more findable, accessible, interoperable, and re-usable, but also about solving the social, cultural, and moral issues that currently limit participation and representation, and prioritize knowledge produced in the global North [6,26,28]. To date, the legacies of colonialism continue to shape debates and practices around open scholarship [29,30]. To address these inequities, open scholarship needs to be understood in relation to the histories, socio-cultural contexts, and political powers that have shaped and continue to constrain the philosophical and ethical impulses of openness [26,31–34].

A growing body of literature is calling for the open scholarship movement to focus more on the cumulative advantage of open science for the global North [5,28,35], framed around solutions to the politics of whiteness, and the dismantling of processes and materials supporting knowledge production that are intertwined with the legacies of colonization [6]. Open science holds the promise to make scientific endeavors more inclusive, participatory, understandable, accessible, and re-usable, engaging those from more marginalized geographies and identities to reach a more holistic understanding of the world and the issues affecting a broader range of audiences [28]. Yet, achieving this will require focusing on the four pillars outlined in UNESCO's Recommendation on Open Science: open scientific knowledge, open science infrastructures, and science communication, with special emphasis on its fourth pillar, that of "open engagement of societal actors and open dialogue with other knowledge systems" [3]. Only through acknowledging knowledge pluralities and encouraging researchers to "situate" their understanding of science within highly nuanced, socio-cultural terrains that shape power structures around open scholarship practices can we begin to acknowledge and construct more equitable opportunities for collaboration, intentionally seeking to create "inclusive infrastructures" to avoid reproducing the status quo of research inequalities [35].

## 3. Power Imbalances Influencing Equity and Inclusion

While digitalization has undoubtedly increased the volume of cultural content available online and facilitated broader community uptake, limited attention has been given to the design of digital platforms and tools to redress the power relations and restructure the systems that reinforce over-representation of knowledge produced by dominant groups [5,6,28,36]. Scholars, designers, and promoters of technical tools and platforms, including those working in galleries, libraries, archives, and museums, choose what to accept, describe, catalog, and document [37]. They decide what to include in their institutions' collections, how to categorize and catalog data, how to annotate, adapt, correct, and modify content, which data to record, tag, and link, and what to include and leave out [34,38]. Defining and developing collections, describing and annotating items, and providing infrastructure are all decisions made in view of aims and interests and in terms of history and politics. And while those choices may not be made through a conscious bias, they can support the very oppressive and restrictive systems they seek to overcome [39].

Christen [40], in her analysis of the connections between information, freedom, and openness, suggested that models of digital curation typically involve three processes: (1) getting/finding, (2) arranging, and (3) sharing content. By simply using a major search engine, we can discover information and images, and download content. Yet, while this digital content may be viewed as open and reusable, it may contain deep ties with the communities, individuals, or groups these materials represent. As such, Christen argues that, in practice, any online search, as well as "data mining", risks being influenced by the colonialist legacies relating to physical materials collected from local places and peoples and grafted onto now-digitized content. Moreover, she argues that metadata—or information about data—can be mired in social, cultural, and political values and norms. For example, in Western settings, authors are legally seen as the sole creator of a work, yet in many Indigenous communities, the notion of a single creator is a difficult concept and is in contrast to the value they place on community production, ancestral creation of stories, or other forms of cultural and artistic content [40].

Similarly, de Oliveira et al., in their study of the inclusion of open science in Latin America [41], outlined two different approaches to the open science movement. The first approach focuses on the principles of "acceleration, efficiency and reproducibility" and advocates for the "standardization" or "homogenization" of scientific practices. This approach, which they argue dominates academia, gives limited attention to the uneven production, distribution, and access to scientific knowledge worldwide as a result of languages, cultures, and power. As such, de Oliveira et al. discuss "publication bias" and "structural bias" faced by global South researchers, and highlight the fees charged to publish in open access (OA), the massive concentration of researchers from the global North on editorial boards [42], predominance of English as an academic lingua franca [43], adoption of policies legitimizing only top-ranked universities and publisher oligopolies [20], and the colonialist legacies permeating scholarly discussions [29,44], all of which make the scientific production and circulation of knowledge from non-Western regions less visible [45]. Ultimately, they caution against open science practices focused on "acceleration, efficiency and reproducibility" of knowledge, arguing that they may result in platform capitalism reinforcing the existing global inequalities in academia. Rather, they propose the need for greater emphasis on the second approach grounded in "participation, social justice, and democratization of knowledge" and provide the example of how Latin America has been enacting its own form of open scholarship for decades [41]. In doing so, they challenge the current open access agenda, suggesting that if it is to become truly inclusive, it will need to address current language, cultural, economic, political, and epistemological differences.

The example of Latin America highlights the importance of regional rather than global initiatives. Here, collaborative efforts have resulted in the distribution and sharing of research through a variety of open repositories, including SciELO, Redalyc, and Latindex, among others. SciELO (Scientific Electronic Library Online) is a bibliographic database, digital library, and cooperative electronic publishing model for open access journals aimed at increasing visibility and access for developing countries. Established in Brazil in 1997, it now provides a portal for accessing journals and publications from 15 Latin American countries, as well as South Africa. Redalyc (Red de Revistas Científicas de América Latina, y El Caribe, España y Portugal) is a similar service aimed at building a scientific information system made up of leading journals of all disciplines edited in Latin America and Iberia. LA Referencia (the Federated Network of Institutional Repositories of Scientific Publications) is supported by countries whose open repositories follow interoperability standards. Importantly, Latin American universities encourage staff to use these networks and institutional repositories rather than paying APCs or BPCs [9]. Yet, despite Latin America having strong research traditions and SciELO being searchable via the Web of Science (WoS), in 2019 less than 5% of these outputs were included in the WoS Journal Citation Reports [46].

Similarly, a recent study by Kanna et al. demonstrates how the open-source publishing platform Open Journal Systems has published over 25,000 journal issues with 5.8 million items from 136 countries, with 79.9% from the global South [47]. Yet, this example also illustrates that, despite wide-ranging geographic, linguistic, and disciplinary diversity, only 1.2% are indexed in the Web of Science and 5.7% in Scopus. Therefore, global South scholars are not well ranked according to privileged Western academic reputation and reward systems, placing a further burden on an already unequal hierarchy of knowledge [41]. If open platforms are to drive greater equity of access to assist humankind in taking full advantage of our digital world, then, as outlined by Kanna et al., there is a need to expand and recalibrate the scale, diversity, and recognition of scholarly knowledge by mapping and indexing information and data across countries, regions, languages, and disciplines [47].

These identified priorities require reviewing not just the publishing environment, but also the inequalities present in open data sharing, open methods, open infrastructure, and open evaluation processes. As Ross-Hellauer et al. point out, data sharing aims for increased citation, transparency, reproducibility, and research quality, reuse, and efficiency [28]. But as they illustrate, the blanket appreciation of open access to data ignores

the inequalities in relation to those less able to participate and access data. The authors highlight that open access to data alone is not enough to guarantee effective re-use of data, as this requires skills, money, and computing power. Open methods and open infrastructure, which include the sharing of code, laboratory notebooks, or preregistering of analyses, are also credited with addressing concerns around the integrity, quality, standardization, transparency, and reproducibility of research [41]. Yet, with major commercial publishing corporations like Elsevier, Wiley, and Springer capturing usage through a host of tools skewed toward interoperability of their own products dominated by academia in the global North, there is a growing call for open science and open scholarship infrastructures to be open-source and community-governed, ensuring data availability remains responsive to community needs [28]. Moreover, while new evaluation processes and alternative bibliometrics, or altmetrics, are providing more open, congruent ways for universities to reshape their assessment of research for societal benefit, many assessment systems still favor the global North [24,48]. The rationalization of open scholarship remains routinely located in the discursive debates of Western academia, rationalized around evolving models for scholarly publishing and unfettered access to the sharing of information and data, rather than the diverse systems of meaning produced by people over time and location in diverse settings. Open scholarship does not operate in a vacuum; it is influenced by social, cultural, and political values and norms [26].

## 4. Framing Open Scholarship for Greater Inclusiveness

The Open and Collaborative Science in Development Network (OCSDNet), a research network involving scientists, development practitioners, and community activists from 26 countries in Latin America, Africa, the Middle East, and Asia, was established in 2014 with the aim of investigating how networked collaboration could address local and global challenges to facilitate a fairer and more open environment for sustainable development. The network centered on challenging 'homogeneous, decontextualized, and dehistoricized definitions' of open science, instead calling for more 'situated' knowledge focused on well-being, development, and collective prosperity [35]. That is, making openness to knowledge understood within its particular history and environment and, as such, identifying who is likely to benefit from the production and circulation of knowledge and who is at risk. To whom does knowledge belong, and who gets to participate in knowledge production processes? The OCSDNet Manifesto outlining the seven principles for open and collaborative science was the outcome of many years of discussion, reflection, research, and negotiation about the core values of a vision for the development of a more inclusive open science [49]. Table 1 summarizes the seven principles for Open and Collaborative Science.

**Table 1.** OCSDNet Principles for Open and Collaborative Science (OCSDNet, 2017).

| Principles | Values |
| --- | --- |
| Principle 1 | Enables a **knowledge commons** where every individual has the means to decide how their knowledge is *governed and managed* to address their needs. |
| Principle 2 | It recognizes **cognitive justice**, the need for *diverse* understandings of knowledge making to co-exist in scientific production. |
| Principle 3 | It practices **situated openness** by addressing the ways in which *context*, *power* and *inequality* condition scientific research. |
| Principle 4 | It advocates for every individual's **right to research** and enables different forms of *participation* at all stages of the research process. |
| Principle 5 | It fosters **equitable collaboration** between scientists and social actors and cultivates *co-creation* and social innovation in society. |
| Principle 6 | It incentivizes **inclusive infrastructures** that empower people of *all abilities* to make, and use accessible open-source technologies. |
| Principle 7 | Strives to use knowledge as a pathway to **sustainable development**, equipping every individual to improve the *well-being* of our society and planet. |

In summary, OCSDNet's principles challenge mainstream narratives of openness focused on acceleration, efficiency, transparency, and productivity—highlighting instead the concept of the knowledge commons, social justice, and inclusion through diversity of participation and the integration of community actors to form collaborative connections across traditional and institutional boundaries in an effort to address regional, context-specific issues in the global South [35]. These principles, moreover, emphasize how existing power structures, hierarchies, and even the cultures of collaboration may unintentionally influence norms and standards around knowledge creation and legitimacy, and suggest the need to foster network-building and information-sharing aimed at the creation of locally relevant, freely accessible, and reusable knowledge, including, for example, the use of local rather than Western norms, such as oral traditions and storytelling. The contextualization and situating of open scholarship are central to encouraging local participation regardless of culture, gender, socioeconomic status, or language, facilitating local capabilities to use, share, and create knowledge. This calls for the inclusion of diverse actors and epistemologies, with the goal of intentionally reconfiguring research methods and practices to address the needs of those who are most often marginalized [35].

Importantly, by recognizing current barriers collaboratively with stakeholders, OCSDNet teams have acknowledged that, in changing their culture and policy through long-term strategies and by scaling up openness for the benefit of those disadvantaged, excluded, or otherwise overlooked, open scholarship has the potential to transform the foundational structures of knowledge creation in new and important ways [19,50]. By exposing power relations and inherent biases and by offering spaces, tools, opportunities, and principles that facilitate opportunities for historically marginalized groups to participate in knowledge production and validate new and existing forms of local knowledge, open scholarship can give rise to the concepts of knowledge commons and social justice [41].

## 5. Fostering Collaborative Networks

Engaging in greater collaboration to reach solutions will require fostering dialogue with multiple stakeholders. The following case studies outline South–South and North–South examples of possible solutions aimed at raising cultural diversity, equity, and inclusion through engendering dialogue to build promising alternatives for open scholarship founded on the principles of openness and justice.

### 5.1. SADiLaR—Making Language Resources Open in the South African Context

The South African Centre for Digital Language Resources (SADiLaR) is a national research infrastructure funded by the Department of Science and Innovation (DSI) of the government of South Africa, and forms part of the broader DSI South African Research Infrastructure Roadmap (SARIR) program. SARIR is a strategic national initiative and framework to facilitate medium-to-long-term planning, implementation, monitoring, and evaluation for the provision of research infrastructures (RIs) necessary for a competitive and sustainable national system of innovation. SADiLaR has the function of creating, managing, and distributing digital (and computational) resources as well as applicable software for all of South Africa's official languages. This is performed, inter alia, in order to stimulate and support research and development in the humanities and social sciences. It is the first and currently the only such center in Africa that is charged with creating and managing digital resources and software to support all of the eleven official languages of South Africa: English, Afrikaans, Sesotho, Setswana, Sesotho sa Leboa, Xitsonga, Tshivenda, Siswati, isiNdebele, isiXhosa, and isiZulu.

SADiLaR has a mandate that is split into two main programs: the digitization program and the digital humanities program. The digitization program involves the creation of relevant text, speech, and multimodal resources for research and the re-intellectualization of South Africa's eleven official languages. This is a challenging task, as re-intellectualization in the South African context means the radical transformation of the capacity and role of Indigenous official African languages in carrying and conveying all forms of knowledge in

all spheres of life using all forms of media [51]. The digital humanities program focuses on developing and cultivating research capacity. SADiLaR works with universities, various linguistic communities, and publishing organizations to harness digital resources in all eleven official languages. The specialization nodes of SADiLaR have harnessed, processed, and curated these digital resources and made them available through an open-source online repository. The SADiLaR repository is host to a corpus portal (of all the eleven official languages) that can be queried online, a multilingual learner corpus of academic texts (SAMuLCAT), a dictionary portal and application, and a grammar portal in development that will be launched soon.

SADiLaR has located the Escalator Project at the center of its digital humanities program. The Escalator is SADiLaR's flagship project. There have been sparse and uncoordinated digital humanities activities in the global South and particularly in South Africa over the past decade or so. As a result, there has been a paucity of computational and digital research skills within the disciplines of the humanities and social sciences. SADiLaR has identified the imperative to initiate a national flagship project that coalesces researchers, students, and practitioners from across these disciplines, as well as the broader academy, and actively links them with computational research areas where learning, sharing of best practices, and co-creation of resources can take place. The main aim is to support the development of an active Community of Practice (CoP) in digital humanities and computational social sciences in South Africa. The idea of a CoP derives from the understanding that this is a platform that brings together hitherto disparate individuals to focus on a shared interest with a view to deepen their knowledge and skills and foster collaboration, growth, and collective advancement.

Within the Escalator project, there are two main activities. The first is a monthly Digital Humanities Colloquium series covering a wide array of topics that are presented by digital humanities (DH) experts from across the globe. The topics discussed mostly pertain to DH techniques or methodologies. Topics on Natural Language Processing and applications are also covered. This monthly online event has created a growing academic community that comes together to share ideas and best practices and problem-solve. SADiLaR has hosted these colloquia since October 2020. The second main activity is the DH-IGNITE events. These are regional events that are hosted by SADiLaR in each of South Africa's nine regions (or provinces) in order to bring together varied disciplines, connect people of different academic backgrounds, and proactively create a CoP in DH and Computational Social Sciences (CSS). There are a variety of activities at these events, and they include presentations, lightning talks, panel discussions, and exhibitions. These events attract students and researchers from across the twenty-six public universities in South Africa, as well as freelance media practitioners, archivists, and librarians.

SADiLaR has initiated a responsive and impactful program that has empowered the under-resourced Indigenous African languages through its digitization program, which has seen the harnessing and creation of datasets in these languages, and has led to the creation of multilingual general and academic corpora. The novel Escalator project has, in the first instance, created a link between DH activities and practices in the global North and global South, through the DH Colloquium series. The second effect of the Escalator project is the skilling (re-skilling and up-skilling) of the humanities and social sciences academic community in the areas of computer science and digital methodologies through the DH-IGNITE events. The creation of such an engaged CoP with experts from varied academic backgrounds is a step towards achieving an "engaged inclusive knowledge society." The focus on creating digital resources for the eleven official languages brings to the fore access to communities that have hitherto been neglected, marginalized, and under-resourced.

*5.2. KSHIP—Digitalization of Minority Literature in India*

India is a multilingual country with twenty-two official languages and eleven written scripts, according to the eighth schedule to the Constitution of India. Each language has its own literature and scholarship. Yet, the state of scholarly publishing infrastructure

in India is precarious, and access to scholarly articles in Indian languages is difficult and challenging [52]. KSHIP—or Knowledge Sharing in Publishing—is an independent publishing center aimed at offering access to scholarly publications in languages other than English. Born from a collaborative commitment toward the development of an open access environment, KSHIP has involved the development of a multilingual Open Access Scholarly Publishing platform focused primarily on areas of the humanities and social sciences with an emphasis on Indian languages. Established and managed by the Indian Institute of Technology (IIT) Indore, KSHIP initially comprised the Multilingual Literature Research Database (MLRD), which focused on three languages with the aim of adding more later. The main objectives of the MLRD project have been to create a relational database that includes citations for Indian research articles in English, Hindi (Devnagari), and Hindi (transliteration); ensure that the meta tags recognize each of these citations in relation to the English meta tags as well as to each other; and collect and collate research data initially in Hindi, Malayalam, Tamil, and Bengali and to extend these to other Indian languages. As a multilingual publishing house, KSHIP has specifically targeted scholarly monographs and translations in Indian languages, inviting scholars to host journals in multiple languages. The database serves as a collaborative and comprehensive hub for Indian literature scholarship, with efforts made for it to be a cooperative and community-based crowd-sourced platform [53].

If archives are repositories of power, databases are the digital pathways to that power, and in the case of literature, they are also the road to canonicity. The awareness of databases as a digital site for research and knowledge production is reflected in the digital transition of bibliographic indexes and the creation of databases across humanities and social sciences areas of research. JSTOR, Project Muse, and MLA International Bibliography are just some examples of relational databases that have, over the years, grown both representatively and in the sheer amount of data that is now available for researchers [52]. It is therefore crucial that literature research in Indian languages is accessible for such research queries, and relational databases are built to cater to them so that they are part of the much larger humanities and literature research ecosystem. Not having a digital presence risks neglecting their presence as part of the conversations around research in literature from India and in the global research network. With the aim of developing a publishing catalog that is dynamic and diverse, KSHIP also has the ambition to publish translations of scholarly works in regional languages that can be used by university students in different parts of the country.

### 5.3. Mukurtu—An Indigenous Content-Centered Managment System

The Mukurtu platform is a community-centered content management system (CMS) designed to enable Indigenous researchers to preserve and disseminate digital heritage in accord with cultural protocols. In response to the worldwide momentum toward de-colonization and reconciliation, Mukurtu constitutes a dynamic digital environmental humanities (DEH) intervention for managing Indigenous communities' data [34,54]. The Mukurtu Wumpurrarni-kari archive was established in 2007 through collaborative dis-cussions with Warumungu community members (Northern Australia). Mukurtu is a Warumungu word meaning 'dilly bag' or a safe keeping place for sacred materials, and as such, Mukurtu was chosen as the name for the archive, which represents a safe keeping place where Warumungu people can share stories, knowledge, and cultural material using their own protocols. The grassroots initiative has empowered communities to curate, man-age, and disseminate their digital heritage in logistically flexible, culturally appropriate, and ethically sensitive ways, in spite of the many obstacles to doing so. Managed by the Center for Digital Scholarship and Curation at Washington State University, Mukurtu is dis-tributed under the terms of the GNU General Public License as a series of commonly used licenses ensuring the freedom to download, run, share, study, and modify software. The development of Mukurtu has been supported to a large extent by the US-based National Endowment for the Humanities, the Institute of Museum and Library Services, the Andrew



W. Mellon Foundation, and the National Science Foundation. Streamlining access to digital collections, Mukurtu facilitates the uploading, modification, and management of editable versions of digital materials rather than long-term preservation master files produced at high-resolution specifications such as TIFF, WAV, DNC, INSK, and DPX files. The platform's focus on lower resolution files reduces users' costs while enabling those with limited Internet availability, especially in remote locations, to access material required for research. Enhancing the production of content while traveling or in field settings, the companion app Mukurtu Mobile can be used with the Mukurtu CMS to align data procurement and research methodologies with community requirements and cultural protocols.

The Mukurtu Showcase features ten projects, one of which is Gather. This illustrative example of the application of Mukurtu innovation is based at the State Library of New South Wales in Sydney, Australia. Gather's objective is to facilitate access to the Library's existing holdings of Aboriginal Australian cultural heritage material. The project's core activities include digitally returning copies of historical photographs, manuscripts, and documents to their owners; engaging with communities to identify individuals, locations, and narratives featured in archival images; and expanding the Library's collections of historical material through the addition of local knowledge from Indigenous stakeholders. In keeping with cultural guidelines, access to some content is restricted. Gather is organized into four categories: Mob; Country and Culture; Languages; and Resistance and Activism. An example of a digital object from Country and Culture is Ngunawal Elder Don Bell's story "Great Dividing Range: A Gurulidj (Bunyip) Story" in the form of a 7-min audio file. The Ngunawal are an Aboriginal people of southern New South Wales and the Australian Capital Territory of Australia, including the towns of Queanbeyan, Yass, Tumut, Boorowa, and Goulburn. Ngunawal people traditionally camped on rises near water sources with access to food but avoided camping directly next to rivers for fear of bunyips (creatures regarded as dwelling in rivers, streams, creeks, swamps, waterholes, and billabongs). The Gather version of Elder Don Bell's story includes a transcription along with a map situating the reader geographically. In the narrative, disobedient boys transgress traditional Law by entering a taboo place where they encounter a monstrous bunyip and flee in terror. The violation of cultural protocol not only suggests the importance of environmental stewardship and respect for natural forces but also encodes biocultural knowledge. The storyteller, for instance, associates gum and box trees with taboo sites, and as the boys flee the bunyip, they come across a women's party gathering yams and bracken roots for bushtucker (6 min and 15 secs). In addition to Gather, Mukurtu projects have been developed by the Chugachmiut people of Alaska, the Passamaquoddy people of northeastern North America, and other Indigenous communities, enhancing principles of Indigenous data sovereignty [55].

*5.4. Climate Stories Project—An Open-Access Participatory Platform on Climate Disruption*

The Climate Stories Project (CSP) is an open-access participatory platform designed for sharing narratives of climate disruption, recognizing the need to place human stories at the center of multifaceted climate debates. Focusing on oral histories, CSP brings affective, human-scale considerations to the abstract, technologized, and oftentimes overwhelming nature of climate change discourse. Contributions include accounts of wildfires, floods, and other natural disasters exacerbated by climate disturbance; firsthand observations of changes in seasonal patterns and cycles; expressions of deep concern for the welfare of families, communities, and cultures; inspiring stories of adaptation to weather extremes and rising sea levels; and narratives of community resilience-building through grassroots campaigns and nonviolent resistance. More than an exclusively archival medium, however, CSP aims to enable musicians and other artists, for example, to draw from climate stories in devising creative responses with the potential to engage wider audiences in the climate debate. The Share a Story tab provides contributors with the opportunity to Share Your Climate Story or Share a Climate Story Interview. Additionally, the Climate Stories tab invites users to explore an interactive map of climate narratives from the United States, Canada, Africa, Europe, Asia/Pacific, and Latin America. A vital component of CSP, fur-

thermore, is ongoing community engagement through workshops on climate storytelling, an ambassadors program training individuals to conduct interviews around the world, and a focus on collaborations between musicians and storytellers creatively reinterpreting the platform's rich multimedia content. Amenable for use in educational and heritage settings, the short interviews are curated from their raw, unedited versions. CSP exemplifies an inclusive, participatory, open data project grappling with the global impacts of climate disturbance through oral histories and the creative interpretation of content.

### 5.5. Indigenous Foods Knowledges Network—An Open Initiative on Indigenous Biocultural Knowledge and Food Sovereignty

The Indigenous Foods Knowledges Network (IFKN), in contrast, offers an example of a collaborative open platform underscoring the value of food security, sovereignty, and equity, in close partnership with Indigenous communities. A limitation of the Climate Stories Project is its prevailing Anglocentric focus on North American climate narratives, with comparatively few contributions from Indigenous communities and members of the Global South. Like Mukurtu, IFKN provides, first and foremost, a framework promoting greater equity and inclusion in open Indigenous biocultural knowledge projects responding to climate change urgencies. As a digital channel connecting Arctic and US-American Southwest Indigenous food knowledge systems, IFKN fosters a network of Indigenous leaders, cultural representatives, and heritage scholars concerned with building community capacity through research into Indigenous food sovereignty and knowledge. The two regions of the Arctic and the US-American Southwest hold in common the challenge of revivifying, sustaining, and adapting food-related knowledge systems in the context of the social and environmental precarities of the present era, and this project responds to the challenge of bringing them together. The IFKN Charter encourages research that engages Indigenous knowledge-making processes, integrates Indigenous values, and bolsters the resilience of Indigenous communities [56]. More precisely, the IFKN promotes Indigenous sovereignty, strives for ethical and equitable research partnerships, develops data in alignment with community values, advances models privileging community-centered approaches and equitable knowledge exchange, recognizes Indigenous languages as vital elements of food sovereignty, demonstrates respect for Indigenous epistemologies, supports Indigenous authority over the research projects impacting them, and remains responsive to the concerns of communities in the Arctic and US-Southwest over seeds, plants, animals, land, air, water, and biocultural assemblages [56].

### 6. Discussion

As our population becomes more digital, many are caught in a cycle of poverty, with the digital divide furthering their lack of engagement and representation [6,36,41]. To facilitate greater social inclusion and equity for all, the use of digital tools and processes requires the collective design of more open and collaborative networks to engage a richer diversity of communities [4,27]. We argue that one approach to achieving this is to implement the seven principles proposed in the OCSDNet Manifesto, which is focused on (i) enabling a knowledge commons where all individuals have the means to decide how their knowledge is governed and managed to address their needs; (ii) recognizing cognitive justice and the need for diverse understandings of knowledge-making to co-exist in scientific production; (iii) practicing situated openness by addressing the ways in which context, power, and inequality condition scientific research; (iv) advocating for each individual's right to research and enabling different forms of participation at all stages of the research process; (v) fostering equitable collaboration between scientists and social actors, and cultivating co-creation and social innovation in society; (vi) incentivizing inclusive infrastructures that empower people of all abilities to make and use accessible open-source technologies; and (vii) using knowledge as a pathway to sustainable development [35].

Realizing these principles will require collaboration through respectful debate and discussion for the co-creation of meaningful, collective knowledge [35]. Yet barriers to

collaboration exist even within the very networks themselves, aggravated by the diverse multi-disciplinary and multi-language landscape and interests of those from varied Northern and Southern contexts [57]. To address these barriers, the creation of infrastructures and practices should intentionally include the voices, worldviews, languages, and epistemologies that have to date been largely excluded from the open science system [26,40]. Through adopting the seven principles of OCSDNet, the case studies provided in this article have sought to explore the potential of open scholarship through the creation of new tools and frameworks focused on addressing local issues and needs. The first case study illustrates the importance of native languages to address the uneven patterns of access to information in South Africa and to make the scholarly communication infrastructure for the humanities and social sciences more inclusive for local communities. SADiLaR (South African Centre for Digital Language Resources), with its national digitization program systematically creating resources related to all official languages of South Africa, is significantly contributing to promoting and recognizing cultural diversity and intercultural understanding by enabling advanced research on language transmission and adaptation. This has included the importance of digital upskilling programs and the provision of a co-learning and co-creation environment aimed at generating greater connectivity for the sharing of each other's experiences, raising awareness of possibilities afforded by the digital environment, and meeting national needs. By developing the basis for a system for sharing, linking, and storing data in multiple languages and across national/linguistic boundaries, SADiLaR provides an excellent foundation for the open sharing of a depth of local understanding, stimulating greater knowledge exchange and advancement, with clear benefits for the broader scholarly community.

Similarly, databases offer digital pathways for the collection of literature, developed over time, for the recording, preserving, and sharing of histories and the transmitting of knowledge and culture that can play a social, psychological, spiritual, and political role. Yet, their emphasis on the English language is endangering the power balance, putting the emphasis on a smaller group of scientific communities [57]. KSHIP offers an ambitious example of publishing scholarly works in regional languages that can be used by university researchers and students in different parts of the country and, in the case of literature, promoting the presence of conversations around research in literature from India and across the global research network. The KSHIP platform serves as a collaborative space for Indian literature scholarship, engaging the community in the capturing of sources, writing and programming, enriching, and archiving of data, and various forms of dissemination. This has included crowdsourcing, publishing, and the sharing of efforts [53]. By engaging citizens rather than working alone with academics, crowdsourcing can help open up and link resources to enable digital exploration of archival records and collections in the quest to discover, collect, and preserve knowledge that may otherwise be lost, offering a means to promote democratic and innovative approaches to the management and safeguarding of collections [58]. Rather than merely being an instrument to involve citizens in the delivery of better content to end users, crowdsourcing offers a way of enabling users to participate in the collection and archiving of previously 'untapped' external knowledge, expertise, and interest, thereby raising public awareness, transcending geo-political borders and boundaries, and promoting greater equality and democratization of knowledge [5,59,60].

Yet too often, even where efforts have been made to engage local communities in the knowledge infrastructure, these activities have been framed and packaged according to the socio-political views of the academics, technicians, and funding bodies who have formed part of their development [59], with the participation or representation of local groups being 'shallow' and 'tokenistic' rather than transformative [61]. While many online platforms have facilitated greater user uptake and collaboration, limited attention has been given to redressing the disconnect between narratives around the democratization of knowledge, reinforcing the over-representation of information and the styles of presentation by the dominant groups [5,62]. In the case of Indigenous peoples, taking knowledge, cultural practices, and data onto public platforms requires addressing concerns around ownership,

repatriation, and Indigenous sovereignty [34]. Merely transforming data and making it open and accessible does not adequately address the concerns of Indigenous peoples [6]. To the contrary, advocates are calling for Indigenous peoples to be centrally involved in every step of the process of digitalization and the management and governance of their data [6,34].

The example of the Mukurtu platform illustrates the importance of protocols for participation, repatriation, and the creation of a safe keeping place by providing Indigenous peoples with the right to own, control, access, and process data that derive from them and that pertain to their communities, knowledge systems, customs, and territories. This recognition of sovereignty has been a first step in understanding and placing historically uneven relationships on a more equal footing, facilitating the repositioning and reframing of materials through expanded and enriched metadata, customizable categories and vocabularies, and a focus on Indigenous knowledge that may not be wholly reliant on text, but also on oral storytelling, audio, video, and other culturally specific forms [34]. It has included developing data in alignment with community values through community-centered approaches and equitable knowledge exchange. Unlike most content management platforms, Mukurtu has been designed around respectful social relationships, not just records—creating a safe keeping place for 'holding up' relationships and knowledge for the preservation and reinvigoration of traditional knowledge for future generations [54]. While the futures of Indigenous communities have been constrained by political, economic, and social structures that are legacies of violent histories, the Mukurtu platform is seen as a way of intentionally entangling the old and the new, the traditional and the digital, opening up different types of mediated futures [54].

Furthermore, the examples of open access participatory platforms on climate disruption and Indigenous food sovereignty networks illustrate how digital environmental humanities ideas have been implemented in significant projects that engage with, envision, re-imagine, and foster communities for environmental action and transformation. These platforms aim to democratize environmental knowledge through open, community-engaged methods, including recognizing Indigenous languages as vital elements of food sovereignty. The multifaceted, multidisciplinary, community-focused, and often participatory orientation of these platforms aims to broaden public awareness of interlinked cultural and ecological urgencies. Through the use of different methods—including the archival conservation of biocultural heritage and the making of digital artifacts informed by ideas of environmental ethics, ecological justice, ecofeminism, sustainability, bioregionalism, biodiversity conservation, and multispecies thinking—such platforms offer opportunities for expanding participation to a broader community, to grapple with the direct impacts of climate change, ecosystem degradation, and biocultural loss [63]. These examples highlight the importance of linking community and Indigenous peoples' oral histories and stories for the sharing of firsthand accounts of climate change and public domains for food security focused on greater equity and inclusion in biocultural knowledge projects. Along similar lines, the Indigenous Knowledge Weather Project demonstrates the movement of traditional biocultural forms (Aboriginal Australian calendars) into the digital public domain through ongoing collaboration between the Bureau of Meteorology and Indigenous communities [64]. These open and innovative communication practices give an insight into the importance of discussions based on Indigenous knowledge sharing and practices passed across generations, intertwined with recognition of sovereignty and community ownership of communication resources [6]. Cultural diversity therefore has a clear role to play in the design and management of open access infrastructure and open communication, and should embrace access to multilingual, multi-disciplinary, and multi-stakeholder content, allowing communication and participation within and beyond the research community [57].

## 7. Conclusions

This paper has revealed a wide disconnect between the aspirations of the open scholarship movement and its promise for a more equitable democratization of knowledge, and the socio-political histories and realities of well-resourced, privileged institutions, a situation that has served to further exacerbate knowledge and research inequalities. Digital technologies have generated new forms of knowledge production and circulation and new types of access and modes of curation, providing many avenues for the sharing and examining of knowledge by diverse citizens, higher education institutions, research centers, government and non-government organizations, and private institutional and international bodies for the benefit of society [2,18]. The design of trusted digital infrastructure resources, openness to archival data, global access policies, and values embedded in international copyright and IP laws have sought to improve information access globally. Yet, moves toward openness have too often been decontextualized from their historical, political, socio-cultural, and economic origins, widening the knowledge gap between the global North and South and further excluding those more marginalized [29,30]. In other words, despite their clear goals for greater democratization of knowledge, open systems have the potential to replicate the very values the movement has sought to challenge [26]. If we are to address the risk of further widening the power imbalances, then efforts to promote openness must be grounded in their historical and socio-political contexts [33,40].

Today, while digital technologies are creating unprecedented and almost unlimited possibilities to satisfy the demand for a stream of high-quality, audience-specific, tailored digital content, questions remain around the extent to which the benefits are truly democratic and fully inclusive. Effective communication requires a multi-directional flow of ideas between different groups within the community. Without this, information can be siloed around 'knowledge clubs' of exclusive academic or scholarly status, impeding input from more vibrant community spaces for the exchange of knowledge that embraces inclusion [5,57,65]. Core to enabling collaboration is a give-and-take between all groups of stakeholders in the technical design, archiving, production, and management of open platforms, with community participation starting from the project's inception. But this requires investing resources in different forms of technical support and training for each member, as well as time to build relationships [34]. Technology alone will not save or revive languages or cultures, or ensure sovereignty is enacted, but by building awareness of the need for more inclusive and equitable systems of knowledge production and sharing and by empowering people to create their own digital systems for the cultural preservation and production of knowledge, we can begin to reduce the historical and contemporary harms imposed through the traditional academic system, and make scholarship more open, collaborative, and fair.

## 8. Future Directions

Based on the concepts and case studies elaborated in this paper, we recommend a series of actions to build commitment to open scholarship philosophies for more equitable, sustainable, and readily available public utilization of knowledge. These include more reflexive, critical, and just modes of working together to promote common understanding, common values, and the common good. Above all, it requires encouraging those in the global North to analyze their 'open' systems through new lenses, reflecting on their histories, reviewing what negative impacts these may potentially have on those less privileged or those in the global South, and questioning how knowledge infrastructure has been designed and to what extent this is controlling or biasing the type of information included or left out. Openness can give rise to and support the concept of a global knowledge commons for social equity and justice. Yet, to do so requires considering open scholarship at the international scale, socially and culturally, through the development of collaborative North–South and South–South networks and by retaining community ownership of local and Indigenous knowledge [41]. This includes encouraging a constant appraisal of how agendas, standards, and norms are set, and by whom. By working together to build and

learn from successful case studies like those outlined above, we can actively seek to share learning and enrich education more openly and equitably [6].

**Author Contributions:** Conceptualization, P.L.A., L.H., J.C.R., N.M. and L.K.; methodology, P.L.A. and L.H.; validation, J.C.R., N.M. and L.K.; investigation, P.L.A., L.H., J.C.R., N.M. and L.K.; writing—original draft preparation, P.L.A. and L.H.; writing—review and editing, P.L.A., L.H., J.C.R., N.M. and L.K. All authors have read and agreed to the published version of the manuscript.

**Funding:** This research received no external funding.

**Data Availability Statement:** Not Applicable.

**Conflicts of Interest:** The authors declare no conflict of interest.

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
