# Peer review of "Making Open Scholarship More Equitable and Inclusive"

_publications, doi:10.3390/publications11030041_

Round 1
Reviewer 1 Report
In the summary and introduction it is anticipated "It analyzes the origins of the open scholarship movement and explores how systemic factors have influenced, and continue to impact, the equality and equity of knowledge access and production to open resources according to location, nationality, race, age, gender, and socio-economic circumstances. It highlights how the privileges of those in the global North permeate academic and technical standards, norms, and infrastructure” but as the article progresses I miss the methodology used and the results that corroborate the analysis presented. There is no empirical research.
The case studies are relevant and interesting, but it would be necessary to identify which authors have been involved, what role they have played in the project and/or case and go much further than a description.
I consider that the research could be published in a section of experiences or case studies.
Author Response
Dear reviewer,
Thank you for your instructive comments. The paper has been substantially edited and improved throughout, including:
- Clarification of the methodology being focused on case studies of international practice; and
- Sentence-level changes and editing through the paper to bring out key arguments and information.
There have also been changes to limit duplication of material, and sharpen the focus of the paper.
Kind regards, and thank you.
Reviewer 2 Report
This article argues that open science should be more equitable and inclusive in order to improve the problems of open science.
In particular, the authors argue that vested interests centered on the global north are still against the global south.
In order to solve these problems, seven principles of OCSDNet were suggested and cases were reviewed.
However, the following limitations need to be improved.
In Chapter 2, open scholarship was analyzed only after the advent of the Internet, but it is necessary to add a historical discussion about open scholarship before the Internet era.
In addition to the strengths of the four cases laid in Chapter 5, it is also necessary to analyze the challenges and limitations of the cases to be solved.
Overall, the same content is repeatedly expressed, and there are many enumerated sentences, making it difficult to read. These need to be simplified for clarity.
In the case of Reference No. 52, the author's name includes 'forthcoming', and the publication year is missing. It is also necessary to check that other references are properly and accurately written.
Author Response
Dear reviewer,
Thank you for your instructive comments. The paper has been substantially edited and improved throughout, including:
- Clarification of the methodology being focused on case studies of international practice;
- Sentence-level changes and editing through the paper to bring out key arguments and information;
- Addition to Section 2 to acknowledge open scholarship prior to the Internet;
- Mention of the challenges in the case studies, in each case;
- Editing of the text overall to avoid duplication and improve readability; and
- Checking of references, including removal of those 'forthcoming'.
There have also been changes to the text throughout to sharpen the focus of the paper and remove redundant material.
Kind regards, and thank you.
Round 2
Reviewer 1 Report
I reaffirm my first review since the changes produced are minor and do not modify the deficiencies identified. The case studies developed are relevant, but the research does not make contributions but is limited to a mere description.
Reviewer 2 Report
Change the title to "Case studies for making open scholarship more equitable and inclusive."